# Validation of On-Farm Bacteriological Systems for Endometritis Diagnosis in Postpartum Dairy Cows

**DOI:** 10.3390/ani11092695

**Published:** 2021-09-15

**Authors:** Nicolas Barbeau-Grégoire, Alexandre Boyer, Marjolaine Rousseau, Marie-Lou Gauthier, Jocelyn Dubuc

**Affiliations:** 1Faculté de Médecine Vétérinaire, Université de Montréal, St-Hyacinthe, QC J2S 2M2, Canada; nicolas.barbeau-gregoire@umontreal.ca (N.B.-G.); alexandre.boyer.1@umontreal.ca (A.B.); marjolaine.rousseau@umontreal.ca (M.R.); 2Complexe de Diagnostic et d’Épidémiosurveillance Vétérinaires du Québec, Ministère de l’Agriculture, des Pêcheries et de l’Alimentation du Québec, St-Hyacinthe, QC J2S 2M2, Canada; marie-lou.gauthier@mapaq.gouv.qc.ca

**Keywords:** dairy cow, endometritis, diagnosis, bacteriology, cytology

## Abstract

**Simple Summary:**

Endometritis is a disease affecting reproductive performance in dairy cows. Considering the modern issues concerning the use of antibiotics in animal production, it is important to refine our criteria for diagnosing this disease. As such, confirming the presence of bacteria in the uterus before implementing an intrauterine antibiotic treatment is critical. To be able to achieve this on dairy farms, the accuracy of currently available on-farm bacteriological culture systems (Tri-plate and Petrifilm) needs to be validated. This study used data from 189 dairy cows to assess this objective. Uterine samples were collected on cows between 30 and 43 days in milk and were submitted for bacteriological culture using three different approaches: standard laboratory and two on-farm systems (Tri-plate and Petrifilm). Our results showed that the optimal criteria for using the Tri-plate and Petrifilm on-farm systems were >90 and >100 colonies, respectively, when compared with the results from the standard laboratory. These results support the possibility of using the Tri-plate on-farm bacteriological culture system to diagnose endometritis.

**Abstract:**

The objective of this study was to validate the accuracy of the results of on-farm bacteriological culture media (Tri-plate and Petrifilm) from endometrial samples compared with the ones from the diagnostic laboratory. A cross-sectional observational study was set up within two dairy herd clients of the Université de Montréal. A total of 189 cows in the postpartum period were systematically enrolled to collect two uterine samples from cytobrushes during the same examination. The first cytobrush was used to inoculate the Tri-plate medium directly and then was sent to the reference laboratory for aerobic bacterial culture. The second cytobrush was used to make a microscopic smear for cytological analysis (proportion of polymorphonuclear cells) and subsequently diluted in 1 mL of saline to inoculate the Petrifilm medium. From these data, statistical analyses were computed to optimize the summation of sensitivity and specificity of the two systems compared with the results of the reference laboratory. For the Tri-plate and Petrifilm media, the cutoffs of ˃90 and ˃100 colonies gave the maximum sum of sensitivity and specificity, respectively. In conclusion, Tri-plate media was best at reproducing the results obtained by laboratory analysis using a threshold of >90 colonies.

## 1. Introduction

Endometritis is defined as the presence of localized inflammation in superficial layers of the uterine body (endometrium) at ≥21 days postpartum [1,2]. Even if this condition does not affect the general health status of cows, it can still have a significant impact on subsequent reproductive performance. Endometritis has been shown to increase the number of days open as well as to decrease the conception rate at first breeding. The magnitude of these effects depends on the severity of the inflammation found in the endometrium of cows [1,3,4,5].

Endometritis is generally diagnosed based on the inflammation level at which the cow’s subsequent fertility will be negatively affected [6]. However, different approaches have been developed over the last 20 years to diagnose endometritis [3,7]. For example, diagnosis of clinical endometritis is based on the presence of clinical signs, like uterine horn or cervix size, as well as a classification of intravaginal discharge. Cytological endometritis is based on the level of inflammation found in the endometrium, like the proportion of polymorphonuclear leukocytes (PMNL) found in uterine body wall or the leukocyte esterase test result. Most researchers have agreed that diagnosing cytological endometritis is the most accurate way to determine the endometritis status of dairy cows, although it is known that cows with purulent vaginal discharge do not always have cytological endometritis [1,7].

The use of intrauterine infusion of cephapirin has been shown to have a good efficacy to treat endometritis and is commonly used on farms [8,9,10,11]. However, the use of antibiotics on farms is likely to evolve over the next decade. Considering the new information available on antibiotic resistance, veterinarians keep adapting their antibiotic use to be as rational as possible. With this idea in mind, confirming the presence of pathogens causing a disease is becoming more and more common before making a decision to use antibiotics. A good example of this situation is how selective dry-cow antibiotic therapy has become common in recent years and will likely replace systematic therapy over time [12]. In such a context, performing a bacteriological culture test on milk samples before dry-off to determine if bacterial pathogens are present will guide the decision of whether to administer antibiotics at dry-off. Practical on-farm bacterial culture systems such as Petrifilm (3M, London, ON, Canada) and Tri-plate (University of Minnesota, MN, USA) can even be used with great accuracy by veterinarians and farmers in order to implement selective dry cow therapy [12,13,14].

Confirming the presence of bacterial pathogens in the uterus of cows would help to avoid excessive use of an intrauterine antibiotic for the treatment of endometritis in cows that do not need it (no bacteria present in uterus). Even though identification of specific bacteria in endometritis cases has already been investigated in some studies [5], little is known about the relationship between the response to antibiotic treatment and the presence of uterine bacteria. Furthermore, there are very limited data available about the diagnostic accuracy of on-farm bacteriological culture systems like Petrifilm and Tri-plate for quantifying the presence of bacterial pathogens in the uteri of cows [15].

The main objective of this study was to quantify and to maximize the accuracy of Petrifilm and Tri-plate on-farm bacterial culture system results compared with those obtained from a standard bacteriological laboratory analysis. We hypothesized that on-farm culture media yield results similar to standard laboratory analysis. A secondary objective of this study was to compare the agreement between cytological and bacteriological results. We hypothesized that cytology diagnoses more positive cases than bacteriology because inflammation could remain active even after infection has been mitigated.

## 2. Materials and Methods

A cross-sectional study was conducted on two commercial dairy farms selected by convenience for being located within one hour of the Bovine Ambulatory Clinic of the Faculté de médecine vétérinaire of the Université de Montréal (St-Hyacinthe, QC, Canada). Both herds had Holstein cows, and the lactating herd size was 200 and 225 cows for herds A and B, respectively. Cows from herd A were housed in a freestall barn, whereas cows from herd B were housed in a tiestall barn.

Data were collected between September 2018 and May 2019. During the project, both farms were visited every 14 days by a research team (a graduate student and an animal health technician). All cows having between 30 and 43 days in milk (DIM) were enrolled in the study. No cows enrolled had received treatment for endometritis before our sampling. The estimated sample size for this study was 200 cows based on finding a sensitivity (Se) and specificity (Sp) of 80% with a minimal acceptable lower confidence limit of 55%, and a prevalence of bacterial contamination of 20% [16].

For each cow enrolled in the study, two samples of the endometrium were collected with cytobrushes. The technique used was adapted from the one described by Kasimanickam [17]. Specifically, the vulva was cleaned with a solution of 2% chlorhexidine gluconate concentrate (Chlorhexidine 2% Solution, Partnar Animal Health, Ilderton, ON, Canada). Subsequently, a first operator inserted a sterile plastic sheath through the cervix of the cow by rectal palpation. A second operator then inserted a sterile cytobrush (Cytobrush Plus GT, CooperSurgical, Trumbull, CT, USA) mounted on a sterile steel rod into the sheath. At this point, the first operator was able to sample endometrial cells by exposing and turning the cytobrush onto the endometrial wall. When the first sample had been collected, the second operator retrieved the first cytobrush and inserted a second one into the plastic sheath that remained inserted through the vagina and cervix by the first operator. The second sample was taken exactly like the first one.

The first cytobrush was used to inoculate the Tri-plate on-farm culture system directly by rolling the brush on the three media. The same brush was then placed in a transportation medium (BBL Port-a-cult Tube, Bioquest, Div. of Becton-Dickinson, Cockeysville, MD, USA) to be sent to the Veterinary Bacteriology Laboratory of the Université de Montréal (St-Hyacinthe, QC, Canada; VBLAB).

The second cytobrush was used to prepare a microscope slide for assessing the PMNL count and to inoculate the Petrifilm on-farm culture system. Specifically, the brush was rolled on a microscope slide at the farm before being put in 1 mL of saline solution (NaCl 0.9%, ICU Medical Canada, Saint-Laurent, QC, Canada) to inoculate the AC Petrifilm plate. Back at our laboratory, both on-farm culture system plates were then incubated for 48 h in standard conditions (37 °C).

After incubation, an observer visually counted the number of colonies on each system based on the technique recommended by the manufacturer. For the Tri-plate system, if there were too many colonies on a medium to be able to count them individually, the medium was given a maximum number of 100. The counts of the three types of culture media (focus, factor and MacConkey) were kept separately. Because the focus and the factor media allowed growth of the same kind of bacteria (gram-positive bacteria), having the separate count results gave us the opportunity to only consider the overall sum of factor and MacConkey media and thus to avoid duplicate counts (focus and factor media). The observer counting the Tri-plate and Petrifilm media was blinded to the other results.

For the VBLAB analysis (reference test), the samples were kept at 4 °C and sent within 24 h of the collection. Standard aerobic bacterial culture was performed for each sample. Specifically, uterine cytobrushes were used to inoculate Columbia agar medium with 5% sheep blood (BD Difco, Fischer Scientific, Ottawa, ON, Canada). The specimens were plated using the streak plate method over four successive quadrants using an aseptic technique. Following inoculation with the cytobrush, a sterile loop was initially dragged from the inoculated section and spread out into a second section. The loop was then dragged from the second section and spread out into the third section, and the steps were repeated for the third and fourth sections, ensuring that sections one and four did not overlap. Plates were then incubated a total of 48 h at 35 °C ± 2 °C with 5% carbon dioxide. Systematically, each sample was also inoculated in a brain heart infusion broth (BD Difco, Fischer Scientific, Ottawa, ON, Canada) that was used as an enrichment method. The importance of growth was classified according to the number of colonies (VBLAB chart): enriched only (bacteria was only identified in the culture broth), rare (one colony grew on the first quadrant), few (2–4 colonies grew on first quadrant), 1+ (≥5 colonies grew on the first quadrant), 2+ (presence of a colony on the second quadrant), 3+ (presence of a colony on the third quadrant) and 4+ (presence of a colony on the fourth quadrant). A sample was considered positive for bacterial endometritis if bacteria were present.

For the cytology analysis, the microscope slides were stained on the day of collection using a Diff Quick kit (Jorgensen Laboratories, Loveland, CO, USA). Once dried, slides were examined under a microscope at 100× to get a general appreciation of the sample. After that, the observer zoomed at 400× to complete a differential cell count of 100 cells focusing on PMNL and endometrial cells. The count was repeated three times at different spots on the slide and an average was calculated. All the slides were read by the same trained observer. A confirmation was completed by an external observer (trained animal health technician) for the slides with a count around 6% (5–7%) of PMNL. This confirmation was completed to ensure that cows were classified in the right group, because the threshold used in this study was ≥6% of PMNL based on previous research [6]. The two observers were both blinded to the results of culture systems at the moment of the slide reading.

All statistical analyses were performed using the program SAS version 9.4 (SAS Institute, Cary, NC, USA). The cow was the research unit of this study. For all statistical analyses, the reference test was the VBLAB. For each on-farm culture system (Tri-plate and Petrifilm), and based on their distributions of the counting results, different dichotomization thresholds using constant intervals were obtained. From 2 × 2 tables (PROC FREQ) based on these thresholds, the performance of the different diagnostic approaches was evaluated considering the following statistics: Se, Sp, positive predictive value (PPV), negative predictive value (NPV) and apparent prevalence (AP). Because our goal was to minimize misclassification (the fewest false positives and false negatives as possible), the highest summation of Se and Sp was considered the reference statistic to acknowledge the best threshold for each on-farm bacterial test. For the two best thresholds based on the highest summation of Se and Sp, the kappa statistic was calculated to quantify the agreement between the VBLAB and the culture system results. To compare the results from the VBLAB and PMNL counts, a 2 × 2 table (PROC FREQ) was computed to obtain agreement beyond chance.

## 3. Results

A total of 203 cows were enrolled in the study. Because of a period of transportation that was too long between sampling and reception at the VBLAB, the samples from 14 cows were not correctly preserved. For that reason, the results of 189 cows (herd A, *n* = 93; herd B, *n* = 96) were used for statistical analyses. Based on the VBLAB results, the prevalence of bacterial endometritis was 16.4% (*n* = 31).

### 3.1. Petrifilm System

The Petrifilm system results are presented in Table 1. The apparent prevalence when using different thresholds varied from 73.0% (*n* = 138; threshold ˃20 colonies) to 17.5% (*n* = 33; threshold ˃200 colonies). The highest summation of Se and Sp was reached when using the threshold of ˃100 colonies, with an Se and Sp of 56.7% and 72.3%, respectively. The kappa statistic using this threshold was 0.20. At this threshold, the PPV and NPV were 27.9% and 89.8%, respectively.

### 3.2. Tri-Plate System

The Tri-plate culture system results are shown in Table 2. The apparent prevalence varied from 14.8% (*n* = 28; threshold ˃100 colonies) to 52.9% (*n* = 100; threshold ˃10 colonies). The highest summation of Se and Sp was reached at the threshold of ˃ 90 colonies, with an Se and Sp of 73.3% and 94.3%, respectively. Using this threshold, the kappa statistic was 0.67. At this threshold, the PPV and NPV were 71.0% and 94.9%, respectively.

### 3.3. VBLAB Results and PMNL Count

The distribution of cows based on their cytology and bacteriology (VBLAB) results is presented in Figure 1. A total of 70 cows were considered positive for endometritis based on cytology (≥6% of PMNL). This represents an apparent prevalence of 37.0% in our sampling group. The number of positive cases was 31 (16.4%) using bacteriology. Overall, 80.6% (25/31) of bacteriology-positive cases were also positive based on cytology. On the other hand, 64.3% (45/70) of positive cases based on cytology were negative based on the bacterial approach.

## 4. Discussion

### 4.1. On-Farm Bacteriological Systems

To our knowledge, this is the first study designed to compare results from a standard bacteriological laboratory analysis with an on-farm bacteriological culture growth when using uterine samples. The optimal thresholds obtained to maximize the total sum of Se and Sp, and hence to minimize the number of misclassifications, were ˃90 and ˃100 colonies for the Tri-plate and Petrifilm systems, respectively. The proportion of cows that were positive for bacteriology based on the reference test (VBLAB) was 16%, while it was 16% and 32% using the aforementioned thresholds for Tri-plate and Petrifilm culture systems, respectively. Thus, the use of the Tri-plate system provides a similar proportion of positive cows than the reference test whereas the use of the Petrifilm system doubles it.

The agreement between the Petrifilm results and the VBLAB results (reference test) was low (kappa = 0.20). Our results for the Petrifilm medium (Table 1) showed that as the threshold increased, the gain in Sp was too small to compensate for the loss in Se. Thus, it globally did not improve the sum of Se and Sp a lot. That observation indicates that in our study population, many positive and negative cows (based on the VBLAB results) had comparable colony counts. Considering that on-farm contamination might have been present, it is possible that the Petrifilm medium was just too sensitive for the growth of bacteria in our farm data collection situation. Perhaps the background bacterial contamination was too high for a medium like Petrifilm. However, this result does not corroborate with the application used in another study [18]. They used Petrifilm media (aerobic and enterobacteria) with a threshold of 5 colonies (≥5 colonies was considered a positive case) to perform a selective antibiotic treatment on cows considered to have endometritis based on an abnormal vaginal discharge (≥1 score of vaginal discharge). It has to be noted that in their samples, only cows previously tested positive for vaginal discharge were then tested with Petrifilm, which could imply that our study population was different. The uterine swabs were also diluted in 3 mL of lysogenic broth medium compared with 1 mL of saline solution in our study. Nevertheless, in the future researchers should consider this difference when building their approach.

For the Tri-plate medium, accuracy compared with the VBLAB results (reference test) was high (kappa = 0.67). Our results showed that the sensitivity did not decrease much between the thresholds 20 and 90 (it stayed at 73%). We can then assume that no positive cows based on the VBLAB results had a colony count between 20 and 90. This observation suggests that most of the positive cows based on the results obtained by the diagnostic laboratory had a colony count <20 and that the rest were much higher (over 90).

Considering Se, Sp and the agreement results (based on the kappa statistic with the VBLAB results), the results of this study suggest the use of the Tri-plate medium for on-farm bacteriological culture of uterine swabs is numerically better than the Petrifilm medium when compared to the VBLAB results. However, this research project and its sample size estimation were not designed to compare the Tri-plate and Petrifilm media directly. Future studies should consider looking at subsequent reproductive performance of cows diagnosed with these tests.

These results should be interpreted with the consideration that no diagnosis of clinical endometritis was performed in the present study. Discriminating cows with clinical endometritis from cows with cytological endometritis might have yielded different Se and Sp results but it was not feasible to explore it in the present study. Another point to consider when interpreting these data is that no identification of specific bacteria were performed; only total aerobic counts were used. This was consistent with the fact that on-farm culture systems used in the study (Tri-plate and Petrifilm) do not allow for such identification. However, it has been shown that not all bacteria have the same pathogenicity for causing endometritis, and potentially some bacteria identified in our counts might not be pathogenic [19,20].

### 4.2. VBLAB Results and PMNL Count Comparison

A secondary objective of our study was to quantify the agreement between cytological and bacteriological results. Of all the cows that tested positive for bacteria, a very large majority of cows were also positive for cytological diagnostic criteria. In the six cows that were positive for bacteria but negative for cytology, their PMNL proportion were all between 3% and 5% (diagnostic criteria for cytological endometritis was ≥6%). This was too low to be considered positive for cytological endometritis based on the a priori threshold used in the study. This relationship between bacteria and cytology results would have been different if another PMNL proportion threshold had been used in our study. This should be kept in mind when interpreting our data.

About two thirds of the cows that were positive for cytology were negative for bacterial criteria. This is important information to keep in mind. From a practical point of view, it implies that a large proportion of cases diagnosed with cytology (as commonly done on farms currently) did not have bacteria that could be found using traditional bacteriological methods. The presence of inflammation without bacteria might be due to residual inflammation (high inflammation after the elimination of the bacterial contamination). However, it can be speculated that the disagreement between the two approaches is not only due to this factor. When planning the study, we made an arbitrary choice to use only traditional aerobic bacteriological culture media (VBLAB, Petrifilm and Tri-plate) to be consistent with the fact that such a system would need to be applicable and achievable on a farm. Because anaerobic bacteria could be present in the uterine body (such as *Fusobacterium necrophorum*, *Prevotella melaninogenica* and *Bacteroides* spp.), it is possible that the inflammation observed in our cows was caused in part by those bacteria [19,21]. We also have to keep in mind that competition between bacteria to grow on culture medium is present and that not all the bacteria have the capacity to grow on them. This could also limit our ability to estimate properly the real bacterial population at the moment of sampling. These factors have to be considered when interpreting the results of this study. If the main goal of a research project is to identify all bacteria present at the time of sampling, genomic analysis should be considered. Our study results also suggest that the association between inflammation and the presence of bacteria in the uterine body postpartum should be investigated further. The methods used in this research did not allow for identification of all possible bacteria present at the moment of the data collection. Techniques combining traditional culture methods and polymerase chain reaction analysis for the more fastidious organisms or metagenomics analysis could have given a more complete image. It was not possible to use them due to budgetary restrictions but future studies should consider using them. Nonetheless, our protocol showed the relevance of on-farm bacteria growth from uterine swabs in a way that could be used in day-to-day herd management routines.

## 5. Conclusions

In this study, we quantified the accuracy of the Tri-plate culture system compared with the reference laboratory analysis results when using the optimal threshold of ˃90 colonies; Se and Sp of 73.3% and 94.3% were obtained, respectively. The Petrifilm system optimal threshold to maximize Se and Se was obtained at ˃100 colonies, and obtained values were 56.7% and 72.3%, respectively. These results support the possibility of using Tri-plate on-farm bacteriological culture system to diagnose endometritis.

Comparison of the bacteriological results with the cytological results showed a low agreement. Our data support the possibility that diagnosis based on cytology might overestimate the number of cases in which antibiotic treatment could be used. However, our results are only descriptive and deserve further research to better understand them.

## Figures and Tables

**Figure 1 animals-11-02695-f001:**
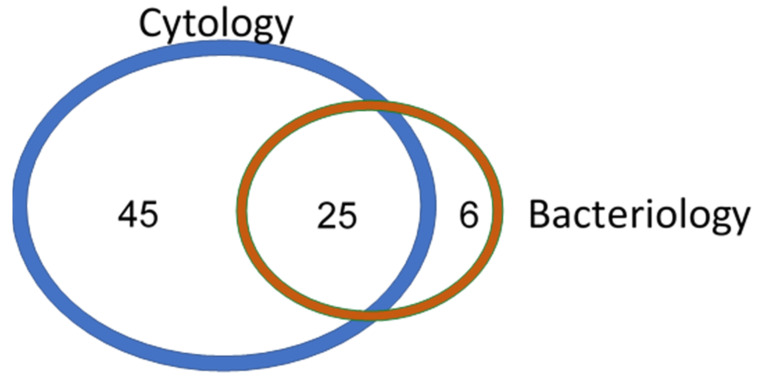
Venn diagram of the distribution of cows based on their cytology (≥6% of PMNL) and bacteriology results (VBLAB; reference test).

**Table 1 animals-11-02695-t001:** Diagnostic performance of the Petrifilm on-farm culture system based on colony count thresholds in comparison with the reference test (VBLAB; *n* = 189).

Threshold (Colony Count)	Sensitivity (%) [95% CI]	Specificity (%) [95% CI]	Sensitivity + Specificity (%)	Positive Predictive Value (%)	Negative Predictive Value (%)	Apparent Prevalence (%)
˃20	83.3 [65.3;94.4]	28.9 [22.0;36.6]	112.3	18.1	90.2	73.0
˃40	63.3 [43.9;80.1]	48.4 [40.4;56.5]	111.8	18.8	87.5	53.4
˃60	60.0 [40.6;77.3]	61.0 [53.0;68.6]	121.0	22.5	89.0	42.3
˃80	56.7 [37.4;74.5]	68.6 [60.7;75.7]	125.2	25.4	89.3	35.4
˃**100 ***	**56.7 [37.4;74.5]**	**72.3 [64.7;79.1]**	**129.0**	**27.9**	**89.8**	**32.2**
˃120	50.0 [31.3;68.7]	76.7 [69.4;83.1]	126.7	28.8	89.1	27.5
˃140	46.7 [28.3;65.7]	80.5 [73.5;86.4]	127.2	31.1	88.9	23.8
˃160	43.3 [25.5;62.6]	83.0 [76.3;88.5]	126.4	32.5	88.6	21.2
˃180	43.3 [25.5;62.6]	84.9 [78.4;90.1]	128.2	35.1	88.8	19.6
˃200	40.0 [22.7;59.4]	86.8 [80.5;91.6]	126.8	36.4	88.5	17.5

* Optimal threshold based on the maximal sum of sensitivity and specificity.

**Table 2 animals-11-02695-t002:** Diagnostic performance of the Tri-plate on-farm culture system based on different colony count thresholds in comparison with the reference laboratory (*n* = 189).

Threshold (Colony Count)	Sensitivity (%) [95% CI]	Specificity (%) [95% CI]	Sensitivity + Specificity (%)	Positive Predictive Value (%)	Negative Predictive Value (%)	Apparent Prevalence (%)
˃10	76.7 [57.7;90.1]	51.6 [43.5;59.6]	128.2	23.0	92.1	52.9
˃20	73.3 [54.1;87.7]	73.0 [65.4;79.7]	146.3	33.8	93.5	34.4
˃30	73.3 [54.1;87.7]	82.4 [75.6;88.0]	155.7	44.0	94.2	26.5
˃40	73.3 [54.1;87.7]	86.2 [79.8;91.1]	159.5	50.0	94.5	23.3
˃50	73.3 [54.1;87.7]	89.9 [84.2;94.1]	163.3	57.9	94.7	20.1
˃60	73.3 [54.1;87.7]	90.6 [84.9;94.6]	163.9	59.5	94.7	19.6
˃70	73.3 [54.1;87.7]	91.8 [86.4;95.6]	165.2	62.9	94.8	18.5
˃80	73.3 [54.1;87.7]	92.5 [87.2;96.0]	165.8	64.7	94.8	18.0
˃**90 ***	**73.3 [54.1;87.7]**	**94.3 [87.2;96.0]**	**167.7**	**71.0**	**94.9**	**16.4**
˃100	70.0 [50.6;85.3]	95.6 [91.1;98.2]	165.6	75.0	94.4	14.8

* Optimal threshold based on the maximal sum of sensitivity and specificity.

## Data Availability

Data are available for consultation by contacting the author of correspondence.

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
