# Peer review of "Validation of On-Farm Bacteriological Systems for Endometritis Diagnosis in Postpartum Dairy Cows"

_animals, 2021, doi:10.3390/ani11092695_

Round 1
Reviewer 1 Report
The submitted revised manuscript with the title “Validation of on-farm bacteriological systems for endometritis diagnosis in postpartum dairy cows” by Barbeau-Grégoire et al. is an improved version with providing more information.
However, some clarifications should be done for final acceptance for publication.
Minor points:
Abstract:
P1/L20-22: I do not agree that the Petrifilm-assay showed sufficient results. Therefore, only the Tri-plate system as a a validated method should be mentioned.
P1/L38-41: The conclusion is still not correct. The Petrifilm system has not a high sensitivity and specificity compared to the Tri-Plate system. Therefore, the conclusion has to be rephrased.
Results:
P5/L192: It should be mentioned that 31 cases are 16% prevalence to compare it to the PMNL results.
P5/L198+202: Is this the reference test or the reference laboratory each? Please clarify it here.
Discussion:
P9/L290-292: The conclusion is still not correct. The Petrifilm system has not a high sensitivity and specificity compared to the Tri-Plate system. Therefore, the conclusion has to be rephrased.
Author Response
Dear Reviewer. Thank you for your comments. We have addressed them below.
---
The submitted revised manuscript with the title “Validation of on-farm bacteriological systems for endometritis diagnosis in postpartum dairy cows” by Barbeau-Grégoire et al. is an improved version with providing more information. However, some clarifications should be done for final acceptance for publication.
Minor points:
Abstract:
P1/L20-22: I do not agree that the Petrifilm-assay showed sufficient results. Therefore, only the Tri-plate system as a a validated method should be mentioned.
--AU: We modifed the last sentence of the summary (L22-23) accordingly. We also did it for the abstract (L36-38) and in the conclusion section of the manuscript (289-291).
P1/L38-41: The conclusion is still not correct. The Petrifilm system has not a high sensitivity and specificity compared to the Tri-Plate system. Therefore, the conclusion has to be rephrased.
--AU: We modified the last sentence of the abstract accordinly (L36-38)
Results:
P5/L192: It should be mentioned that 31 cases are 16% prevalence to compare it to the PMNL results.
--AU: Added accordingly (L189)
P5/L198+202: Is this the reference test or the reference laboratory each? Please clarify it here.
--AU: Changed as suggested (L192-196). We mentioned what that it is based on the reference test (VBLAB) as described earlier in the M&M. This should clarify it for readers.
Discussion:
P9/L290-292: The conclusion is still not correct. The Petrifilm system has not a high sensitivity and specificity compared to the Tri-Plate system. Therefore, the conclusion has to be rephrased.
--AU: Changed as suggested (L289-291)
Reviewer 2 Report
Despite having changed the abstract, it still exceeds the word limit provided for in the instructions for authors
He indicated that the test reference was missing in the discussion, but he did not include it (L216 and L232)
Author Response
Dear Reviewer, thank you for your comments. Ww addressed it below.
Despite having changed the abstract, it still exceeds the word limit provided for in the instructions for authors
--AU: We apologized for this situation. We thought it was 250 words but it is 200. We modified the abstract accordingly.
He indicated that the test reference was missing in the discussion, but he did not include it (L216 and L232)
--AU: We are sure we understand this suggestion. Our VBLAB analysis was the reference test (description at L127-142). To clarify it for readers, we added a mention about it on L127.
Reviewer 3 Report
No further comments
Author Response
No change made.
This manuscript is a resubmission of an earlier submission. The following is a list of the peer review reports and author responses from that submission.
Round 1
Reviewer 1 Report
The objective was to determine if 2 different on-farm bacteria culture systems could be used to identify aerobic bacterial uterine infections in postpartum cows. A secondary objective was to compare the agreement between cytological endometritis as diagnosed by PMN counts and the presence of aerobic bacterial uterine infections.
The study apports useful data and was well designed to resolve the main objective but is only declarative for the secondary objective (no statistical tests were performed).
Main concerns:
Materials and Methods
No rationale behind the selection of the minimal threshold colony count and intervals between thresholds is presented; useful data to be included.
Discussion
The results of the study are only barely contrasted or interpreted according to the results from other authors or studies from the same group; there are only 3 references mentioned throughout the discussion. Must be improved.
Conclusions
Main objective; the conclusion needs to clearly state the usefulness of the tested on-farm bacteria culture systems, according to the objective and presented results (both on the abstract and on the complete text).
Secondary objective; the conclusion must acknowledge the scientific weakness of the presented results.
Specific comments:
Confirm if threshold levels are >than or ≥than.
L 185 and L 203
The number of positive cases using bacteriology does not agree, 30 vs 31 (if necessary recalculate percentage).
L 241-244
Confusing text; clarify interpretation.
L 245
Considering Se, Se and ....; must say, Considering Se, Sp and ....
Author Response
Dear Reviewer 1, thank you for providing very constructive comments to improve our manuscript. We have addressed all of them (see below; our responses start with --AU: ) and changes in manuscripts are highlighted in yellow.
---
The objective was to determine if 2 different on-farm bacteria culture systems could be used to identify aerobic bacterial uterine infections in postpartum cows. A secondary objective was to compare the agreement between cytological endometritis as diagnosed by PMN counts and the presence of aerobic bacterial uterine infections. The study apports useful data and was well designed to resolve the main objective but is only declarative for the secondary objective (no statistical tests were performed).
Main concerns:
Materials and Methods
No rationale behind the selection of the minimal threshold colony count and intervals between thresholds is presented; useful data to be included.
--AU: This is a good point. Because of the evolution of the manuscript over time (before submission) data analyses have evolved. With your comment, we realized that the threshold used in final data analysis was not consistent between analysis and what was presented in manuscript. It was any presence of bacteria (instead of 2+). We adjusted the manuscript (L145) so that now the M&M and the results are coherent. It should answer your concern. Sorry for forgetting to change it before publication.
Discussion
The results of the study are only barely contrasted or interpreted according to the results from other authors or studies from the same group; there are only 3 references mentioned throughout the discussion. Must be improved.
--AU: Good point. We had internal discussions about this before submitting the manuscript initially. The challenge that we face in this situation is that there are almost no data available and published about the comparison of Tri-Plate/Petrifilm results with standard laboratory results. It explains why there are not much references to discuss in the discussion section. Any suggestions are welcome for this point. Being in a field mostly unexplored puts us in unconventional situation when writing a paper. However, we don't believe it misleads readers about available work on this topic or affects the scientific quality of the manuscript. Therefore, we did not change it. We are happy to reconsider if suggestions are made.
Conclusions
Main objective; the conclusion needs to clearly state the usefulness of the tested on-farm bacteria culture systems, according to the objective and presented results (both on the abstract and on the complete text).
--AU: Good point. We added a sentence in the conclusion section (L291-292) to clarify it.
Secondary objective; the conclusion must acknowledge the scientific weakness of the presented results.
--AU: Good point. We added a sentence (L295-296) to clarify it to avoid misleading readers.
Specific comments:
Confirm if threshold levels are >than or ≥than.
--AU: Data were anayzed using > than. We validated that the manuscript was written accordingly and it was ok.
L 185 and L 203 The number of positive cases using bacteriology does not agree, 30 vs 31 (if necessary recalculate percentage).
--AU: Good point. The good number is 31. So we adjusted the manuscript at L175 to clarify it. Thank you for catching this.
L 241-244 Confusing text; clarify interpretation.
--AU: Good point. It was rephrased an expended to clarify for readers (L254-261).
L 245 Considering Se, Se and ....; must say, Considering Se, Sp and ....
--AU: good point. It was changed as suggested (L238). Thank you for catching it.
Reviewer 2 Report
The summary has more than 250 words allowed by the journal, it is necessary to make a shorter summary.
The text of materials and methods is a little confusing. Separate step by step how the test was carried out so that it is clearer how the test was carried out.
Should reinforce your results increase the robustness statistic of your results.
In your discussion you should compare the results obtained with other works and/or justify your statements with references. During the entire discussion, only 3 authors were referenced.
The references are not in accordance with what is requested by the Journal, you must respect what is requested, for example, to Journal Articles:
1. Author 1, A.B.; Author 2, C.D. Title of the article. Abbreviated Journal Name Year, Volume, page range.
Author Response
Dear Reviewer 2, thank you for providing very constructive comments to improve our manuscript. We have addressed all of them (see below; our responses start with --AU: ) and changes in manuscripts are highlighted in yellow.
---
The summary has more than 250 words allowed by the journal, it is necessary to make a shorter summary.
--AU: Good point. This was reviewed and the count is now 250 (L25-41)
The text of materials and methods is a little confusing. Separate step by step how the test was carried out so that it is clearer how the test was carried out.
--AU: Good point. The text was separated in multiple smaller paragraphs to show step-by-step all the procedure and avoid confusion for readers (L96-129)
Should reinforce your results increase the robustness statistic of your results.
--AU: Sorry we don't understand exactly this comment. We verified to ensure that the 95%CI of Se and Sp are presented in the manuscript (L199-204) and clarified in the conclusion that the objective 2 was only descriptive (as requested by Reviewer 1; L295-296). We are open to other suggestions if needed.
In your discussion you should compare the results obtained with other works and/or justify your statements with references. During the entire discussion, only 3 authors were referenced.
--AU: This point was raised by Reviewer 1 as well. We had internal discussions about this before submitting the manuscript. The challenge that we face in this situation is that there are almost no data available and published about the comparison of Tri-Plate/Petrifilm results with standard laboratory results (main objective of the study). It explains why there are not much references to discuss in the discussion section. Any suggestions are welcomed for this point. Being in a field mostly unexplored puts us in unconventional situation when writing a paper. However, we don't believe it misleads readers about available work on this topic or affects the scientific quality of the manuscript. Therefore, we did not change it. We are happy to reconsider if suggestions are made.
The references are not in accordance with what is requested by the Journal, you must respect what is requested, for example, to Journal Articles:
1. Author 1, A.B.; Author 2, C.D. Title of the article. Abbreviated Journal Name Year, Volume, page range.
--AU: Thank you for catching this. We adjusted it as requested. (L321-358)
Reviewer 3 Report
The submitted manuscript with the title “Validation of on-farm bacteriological systems for endometritis diagnosis in postpartum dairy cows” by Barbeau-Grégoire et al. attracted first my attention. The authors collected uterine samples from 189 cows between 30-43 days in milk (DIM). These samples were subjected for bacteriological analysis and cytological analysis. The goal was to validate on-farm bacteriological systems. Statistical analysis were performed to optimize sensitivity and specificity of these tests compared to a standard laboratory analysis. The used Tri-plate system reached a high sensitivity and specificity compared to laboratory. However, the number of samples containing countable bacteria (16 % prevalence) was much lower that the number of samples with more than 6% PMN (30% prevalence).
In general, research improving the diagnosis of endometritis on the farm is very important to improve treatment.
The manuscript is well-written and well-organized.
However, the manuscript shows essential weaknesses in the statement of the aims, the presentation and interpretation of the data.
Therefore, the manuscript in its present form is not in a suitable form for considering for publication.
Major points:
- One weakness of this manuscript is that the authors mixed up cows with signs of clinical endometritis and subclinical/cytological endometritis. The cows should be classified in these two groups because the grade of inflammation and damage of the endometrium is different. In this case, a higher percentage of sensitivity and specificity would be obtained. The number of cows with vaginal discharge is required to state in this manuscript.
- I am not convinced of the stated aim of this study. The signs of clinical endometritis can easily detected by presence of vaginal discharge and the PMN number can be counted in short time for detection of subclinical endometritis. Therefore, a bacteriological test which needs three days is not necessary and it is not useful and practicable. On the other hand, the authors stated on P7/L292-294, that a cytological positive test would overestimate an antibiotic treatment. This is in my eyes would be a reasonable aim of the performed study. The statement of the aims should be rephrased.
- One weakness of this manuscript is that the authors only counted bacterial colonies without considering the pathogenicity of the obtained colonies. This has to be addressed at least in the discussion. There are a lot of known bacterial strains, which not caused the disease endometritis.
- In the same context, the authors should mention that 16 % prevalence of the results of the lab was obtained for the bacteriology. This is nearly the same amount obtained with the use of the Tri-plate system. This should be addressed to the discussion.
Specific points:
Abstract:
- P1/L20-21: This is not correct because the authors only counted the PMN and seems that there were no data presented about vaginal discharge.
- P1/L21-24 +39-41: The conclusion is not correct. The Petrifilm system has not a high sensitivity and specificity compared to the Tri-Plate system. Therefore, the conclusion has to be rephrased.
- The presentation of the obtained results is quite short. It should be mentioned the sensitivity and specificity of the tests. Otherwise, the methods section at this position must be shortened when word-limited.
Introduction:
- P2/L59-60: Cows with vaginal discharge can have a percentage of less than 5 % PMN in the cytological sample. This has to be considered.
Material and methods:
- Please remove “Inc.” and “Co..”from the company names.
- P3/L115: Please introduce here a new paragraph.
- Were for the Tri-plate system only the colonies from focus or factor medium counted and added to the values of the MacConkey medium. This should be explained at the position P3/l129-133. I suggest to check if only counting the colonies on one of these agars would increase the sensitivity and specificity? I think this is an interesting point.
Results:
- P4/L181-183: This information can be removed because the relevant number of cows included in this study is 189.
- P5/L211+213: Is this the reference test or the reference laboratory each?
Author Response
Dear Reviewer 3, thank you for providing very constructive comments to improve our manuscript. We have addressed all of them (see below; our responses start with --AU: ) and changes in manuscripts are highlighted in yellow.
---
The submitted manuscript with the title “Validation of on-farm bacteriological systems for endometritis diagnosis in postpartum dairy cows” by Barbeau-Grégoire et al. attracted first my attention. The authors collected uterine samples from 189 cows between 30-43 days in milk (DIM). These samples were subjected for bacteriological analysis and cytological analysis. The goal was to validate on-farm bacteriological systems. Statistical analysis were performed to optimize sensitivity and specificity of these tests compared to a standard laboratory analysis. The used Tri-plate system reached a high sensitivity and specificity compared to laboratory. However, the number of samples containing countable bacteria (16 % prevalence) was much lower that the number of samples with more than 6% PMN (30% prevalence).
In general, research improving the diagnosis of endometritis on the farm is very important to improve treatment. The manuscript is well-written and well-organized.
However, the manuscript shows essential weaknesses in the statement of the aims, the presentation and interpretation of the data.
Therefore, the manuscript in its present form is not in a suitable form for considering for publication.
Major points:
- One weakness of this manuscript is that the authors mixed up cows with signs of clinical endometritis and subclinical/cytological endometritis. The cows should be classified in these two groups because the grade of inflammation and damage of the endometrium is different. In this case, a higher percentage of sensitivity and specificity would be obtained. The number of cows with vaginal discharge is required to state in this manuscript.
--AU: Very good point. Unfortunately, we did not collect the information about clinical endometritis in this study. We only examined cows at 30-43 DIM with the cytobrushes (not looking if they had purulent vaginal discharge or not). At the time of the start of the study, including diagnosis of PVD was on our mind but we were in an exploration mode (no other data available in the literature at that time) and were did not want to have a large study that would compromise it completion (it was a pilot study to be honest). Knowing what we know today, Reviewer 3 is correct that it should be the next step to discriminate cows with PVD. So to go back to your concern, we are not able to add the requested information. However, we added a sentence in the discussion to clarify that the interpretation of our data should be done considering this point (L244-247)
- I am not convinced of the stated aim of this study. The signs of clinical endometritis can easily detected by presence of vaginal discharge and the PMN number can be counted in short time for detection of subclinical endometritis. Therefore, a bacteriological test which needs three days is not necessary and it is not useful and practicable. On the other hand, the authors stated on P7/L292-294, that a cytological positive test would overestimate an antibiotic treatment. This is in my eyes would be a reasonable aim of the performed study. The statement of the aims should be rephrased.
–AU : Good point. We changed a part of the introduction to address it (L75-77).
- One weakness of this manuscript is that the authors only counted bacterial colonies without considering the pathogenicity of the obtained colonies. This has to be addressed at least in the discussion. There are a lot of known bacterial strains, which not caused the disease endometritis.
--AU: Good point. This was addressed in the discussion section by adding two sentences+references to clarify for readers (L247-252)
- In the same context, the authors should mention that 16 % prevalence of the results of the lab was obtained for the bacteriology. This is nearly the same amount obtained with the use of the Tri-plate system. This should be addressed to the discussion.
–AU : Good point. A clarification was added as suggested (L211-215).
Specific points:
Abstract:
- P1/L20-21: This is not correct because the authors only counted the PMN and seems that there were no data presented about vaginal discharge.
--AU: Good point. Sentence deleted as suggested.
- P1/L21-24 +39-41: The conclusion is not correct. The Petrifilm system has not a high sensitivity and specificity compared to the Tri-Plate system. Therefore, the conclusion has to be rephrased.
--AU : Rephrased as suggested (L20-22 and 38-39)
- The presentation of the obtained results is quite short. It should be mentioned the sensitivity and specificity of the tests. Otherwise, the methods section at this position must be shortened when word-limited.
--AU: After consideration of similar comments made by Reviewer 2, we did shorten the abstract to have no more than 250 words. We understand the point here about providing Se + Sp results in the abstract but we were not able to find a way to include it without deleting sentences of the abstracts that would compromise its self-readability. Therefore, we did not included Se + Sp. We don’t believe it affect the clarity for readers.
Introduction:
- P2/L59-60: Cows with vaginal discharge can have a percentage of less than 5 % PMN in the cytological sample. This has to be considered.
--AU : Good point. We added a clarification at L60-61.
Material and methods:
- Please remove “Inc.” and “Co..”from the company names.
--AU: Changed as suggested all over M&M section.
- P3/L115: Please introduce here a new paragraph.
--AU: As also suggested by Reviewer 1, this big paragraph of the M&M was broken down in many smaller ones to facilitate its reading (step-by-step; L96-129).
- Were for the Tri-plate system only the colonies from focus or factor medium counted and added to the values of the MacConkey medium. This should be explained at the position P3/l129-133. I suggest to check if only counting the colonies on one of these agars would increase the sensitivity and specificity? I think this is an interesting point.
--AU : We did clarify it in the M&M section (L126-128). At the start of the study, we wanted to make sure we would capture gram –positive (factor) and gram-negative (MacConkey), but it turned out that only 5 cows had growth on MacConkey during the entire study. Your suggestion is good so we recalculate our Se+Sp by curiosity. However, it did not make much changes because of the very small proportion of cows involved. It might have been different if the proportion had been bigger but we did not have the sample size to capture it.
Results:
- P4/L181-183: This information can be removed because the relevant number of cows included in this study is 189.
--AU: We prefer to keep it because it shows that we targeted 200 cows (sample size estimation) but we did not achieve it for unplanned logistical reasons. Although it brings a few more lines in the results, it avoids justifying later in the discussion section why we did not reached 200 cows as planned. No change was made to manuscript.
- P5/L211+213: Is this the reference test or the reference laboratory each?
--AU: Good point. Clarifications were added throughout the discussion section (as needed).